# Multi-ancestry sleep-by-SNP interaction analysis in 126,926 individuals reveals lipid loci stratified by sleep duration

Raymond Noordam et al.[#]

Both short and long sleep are associated with an adverse lipid profile, likely through different biological pathways. To elucidate the biology of sleep-associated adverse lipid profile, we conduct multi-ancestry genome-wide sleep-SNP interaction analyses on three lipid traits (HDL-c, LDL-c and triglycerides). In the total study sample (discovery + replication) of 126,926 individuals from 5 different ancestry groups, when considering either long or short total sleep time interactions in joint analyses, we identify 49 previously unreported lipid loci, and 10 additional previously unreported lipid loci in a restricted sample of European-ancestry cohorts. In addition, we identify new gene-sleep interactions for known lipid loci such as *LPL* and *PCSK9*. The previously unreported lipid loci have a modest explained variance in lipid levels: most notable, gene-short-sleep interactions explain 4.25% of the variance in triglyceride level. Collectively, these findings contribute to our understanding of the biological mechanisms involved in sleep-associated adverse lipid profiles.

---

[#]A full list of authors and their affiliations appears at the end of the paper.

D yslipidemia is defined as abnormalities in one or more types of lipids, such as high blood LDL-cholesterol (LDL-c) and triglyceride (TG) concentrations and a low HDL-cholesterol (HDL-c) concentration. High LDL-c and TG are well-established modifiable causal risk factors for cardiovascular disease[1–3], and therefore are a primary focus for preventive and therapeutic interventions. Over 300 genetic loci are identified to be associated with blood lipid concentrations[4–10]. Recent studies showed that only 12.3% of the total variance in lipid concentration is explained by common single-nucleotide polymorphisms (SNPs), suggesting additional lipid loci could be uncovered[10]. Some of the unexplained heritability may be due to the presence of gene–environment and gene–gene interactions. Recently, high levels of physical activity were shown to modify the effects of four genetic loci on lipid levels[11], an additional 18 previously unreported lipid loci were identified when considering interactions with high alcohol consumption[12], and 13 previously unreported lipid loci were identified when considering interaction with smoking status[13], suggesting that behavioural factors may interact with genetic loci to influence lipid levels.

Sleep is increasingly recognised as a fundamental behaviour that influences a wide range of physiological processes[14]. A large volume of epidemiological research implicates disturbed sleep in the pathogenesis of atherosclerosis[15], and specifically, both a long and short sleep duration are associated with an adverse blood lipid profile[16–26]. However, it is unknown whether sleep duration modifies genetic risk factors for adverse blood lipid profiles. We hypothesise that short and long habitual sleep duration may modify genetic associations with blood lipid levels. The identification of SNPs involved in such interactions will facilitate our understanding of the biological background of sleep-associated adverse lipid profiles.

We investigate gene–sleep duration interaction effects on blood lipid levels as part of the Gene-Lifestyle Interactions Working Group within the Cohorts for Heart and Aging Research in Genomic Epidemiology (CHARGE) Consortium[27,28]. To permit the detection of both such sleep-duration–SNP interactions and lipid–SNP associations accounting for total sleep duration, a two degree of freedom (2df) test that jointly tests the SNP-main and SNP-interaction effect was applied[29]. Given that there are differences among ancestry groups in sleep behaviours and lipid levels, analysis of data from cohorts of varying ancestries facilitate the discovery of robust interactions between genetic loci and sleep traits. We focus on short total sleep time (STST; defined as the lower 20% of age- and sex-adjusted sleep duration residuals) and long total sleep time (LTST; defined as the upper 20% of age- and sex-adjusted sleep duration residuals) as exposures compared with the remaining individuals in the study population, given that each extreme sleep trait are associated with multiple adverse metabolic and health outcomes[15–26,30–34]. Within this study, we report multi-ancestry sleep-by-SNP interaction analyses for blood lipid levels that successfully identified several previously unreported loci for blood lipid traits.

## Results

**Study population.** Discovery analyses were performed in up to 62,457 individuals (40,041 European-ancestry, 14,908 African-ancestry, 4460 Hispanic-ancestry, 2379 Asian-ancestry and 669 Brazilian/mixed-ancestry individuals) from 21 studies spanning five different ancestry groups (Supplementary Tables 1 and 2; Supplementary Data 1). Of the total discovery analysis, 13,046 (20.9%) individuals were classified as short sleepers and 12,317 (19.7%) individuals as long sleepers. Replication analyses were performed in up to 64,469 individuals (47,612 European-ancestry, 12,578 Hispanic-ancestry, 3133 Asian-ancestry and 1146 African-

ancestry individuals) from 19 studies spanning four different ancestry groups (Supplementary Tables 3 and 4; Supplementary Data 2). Of the total replication analysis, 12,952 (20.1%) individuals were classified as short sleepers and 12,834 (19.9%) individuals as long sleepers.

**Genome-wide SNP–sleep interaction analyses.** An overview of the multi-ancestry analyses process for both STST and LTST is presented in Fig. 1. QQ plots of the combined multi-ancestry and European meta-analysis of the discovery and replication analysis are presented in Supplementary Figs. 1 and 2. Lambda values ranged between 1.023 and 1.055 (trans-ancestry meta-analysis) before the second genomic control and were all 1 after second genomic control correction. In the combined discovery and replication meta-analyses comprising all contributing ancestry groups, we found that many SNPs replicated for the lipid traits ($P_{joint}$ in replication < 0.05 with similar directions of effect as in the discovery analyses and $P_{joint}$ in combined discovery and replication analysis < $5 \times 10^{-8}$). Notably, we replicated 2395 and 2576 SNPs for HDL-c, 2012 and 2074 SNPs for LDL-c, and 2643 and 2734 SNPs for TG in the joint model with LTST and STST, respectively.

Most of the replicated SNPs were mapped to known loci (Supplementary Data 3 and 4). We looked at the 427 known lipid SNPs (Supplementary Data 5), but these did not reveal significant 1df interactions with either LTST or STST. In addition, we identified lead SNPs mapping to previously unreported regions when considering the joint model with potential interaction for either STST or LTST (>1 Mb distance from known locus). Ultimately, in the multi-ancestry analysis, we identified 14 previously unreported loci for HDL-c, 12 for LDL-c and 23 ci for TG ($R^2 < 0.1$; Fig. 2). Of these, seven loci for HDL-c, four loci for LDL-c and seven loci for TG were identified after considering an interaction with LTST (Supplementary Data 6). Furthermore, 7 loci for HDL-c, 8 loci for LDL-c and 16 loci for TG were identified when considering an interaction with STST (Supplementary Data 7). Importantly, none of these loci for the three lipid traits identified through LTST were identified in the analyses with STST, and vice versa. Furthermore, these lipid loci were specific to a single-lipid trait. Regional plots of the previously unreported loci from the multi-ancestry analyses are presented in Supplementary Figs. 3–8. Some of the previously unreported SNPs identified through modelling a short or long sleep duration interaction (1df) also showed suggestive evidence of association with lipid levels in the joint model (2df interaction test). However, this pattern suggested a main effect that appeared once sleep duration was adjusted for rather than an effect due to an interaction between sleep and the SNP (Supplementary Data 6, 7).

Using the R-based VarExp package[35], we calculated the explained variance based on the summary statistics of the combined discovery and replication analysis. Collectively, previously unreported lead lipid SNPs identified with LTST explained 0.97% of the total HDL-c variation, 0.13% of the total LDL-c variation and 1.51% of the total TG variation. In addition, the previously unreported SNPs identified with STST explained 1.00% of the total HDL-c variation, 0.38% of the total LDL-c variation and 4.25% of the total TG variation.

In the analyses restricted to European-ancestry individuals (overview Supplementary Fig. 9), we identified ten additional previously unreported loci (seven with LTST and three with STST; Supplementary Fig. 10), which were not identified in the multi-ancestry analyses. Of these, we identified four loci for HDL-c, two loci for LDL-c and one locus for TG with LTST (Supplementary Data 8). In addition, we identified one locus for HDL-c and two for TG with STST (Supplementary Data 9).

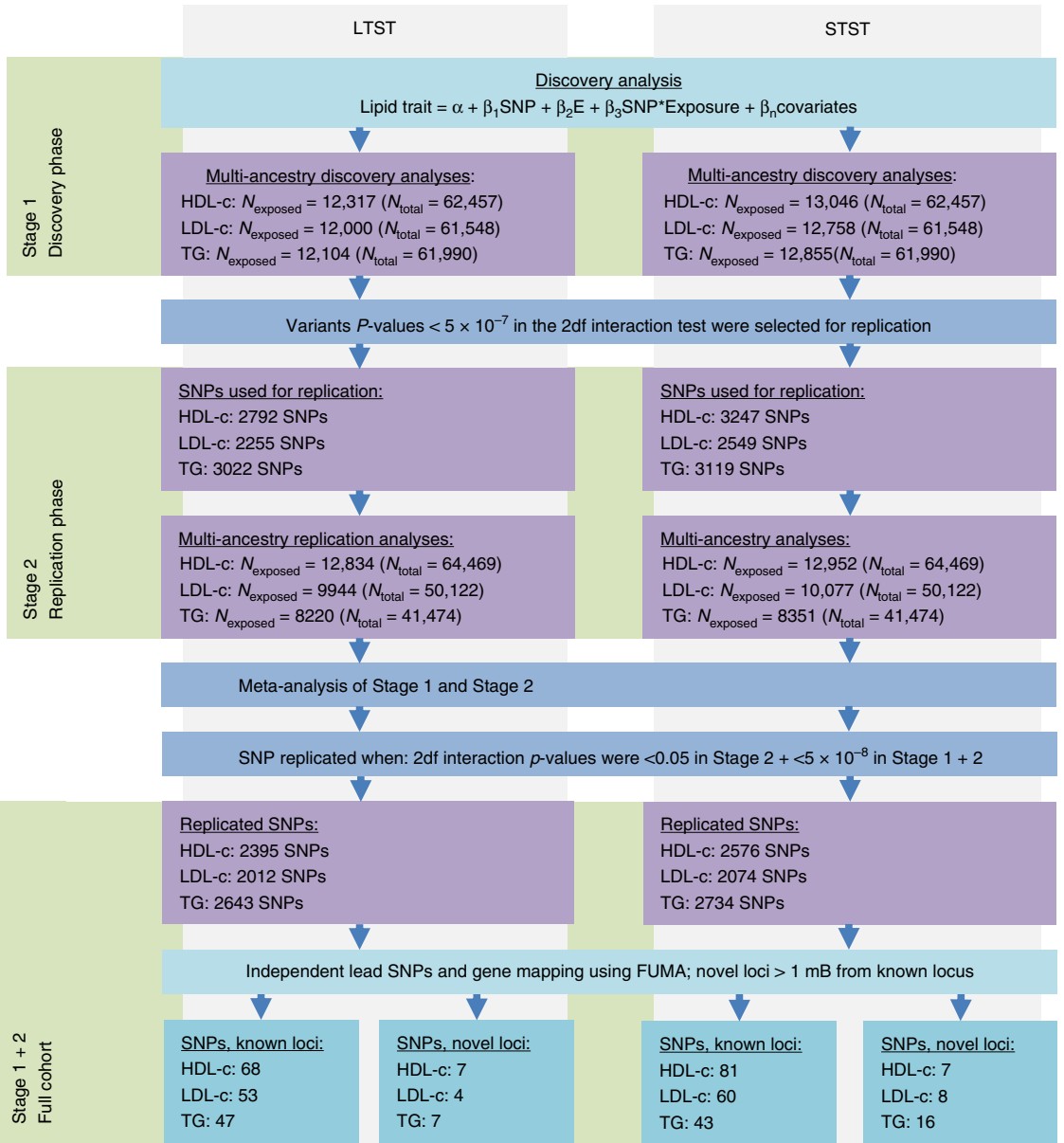

**Fig. 1** Project overview and SNP selection in the multi-ancestry analyses. Project overview of the multi-ancestry analyses of how the new lipid loci were identified in the present project. Replicated variants had to have 2df interaction test $P$-values of Stage $1 < 5 \times 10^{-7}$, Stage $2 < 0.05$ with a similar direction of effect as in the discovery meta-analysis, and Stage $1 + 2 < 5 \times 10^{-8}$

Again, we observed no overlapping findings between the two sleep exposures and the three lipid traits. Regional plots of the previously unreported loci were presented in Supplementary Figs. 11–15.

**Gene mapping of known and previously unreported loci.** Based on a total of 402 lead SNPs in known and previously unreported regions for both exposures and the three lipid traits that were identified using the joint test in the combined sample of discovery and replication studies, we subsequently explored the extent the effects were driven by 1df interaction with the sleep exposure trait being tested[29]. We corrected the 1df interaction $P$-value for multiple testing using the false discovery rate[36] considering all 402 lead SNPs for the present investigation, which was equivalent in our study to a 1df interaction $P$-value $< 5 \times 10^{-4}$. Overall, in the multi-ancestry meta-analyses, the previously unreported lipid loci show clearly stronger interaction with either LTST or STST than the loci defined as known (Fig. 3). The majority of these

identified lead variants were generally common, with minor allele frequencies (MAF) mostly $> 0.2$, and SNP $\times$ sleep interaction effects were not specifically identified in lower frequency SNPs (e.g., MAF $< 0.05$).

Out of the seven previously unreported HDL-c loci identified in the joint model with LTST, six had a 1df interaction $P$-value$_{FDR} < 0.05$, notably lead SNPs mapped to *ATP6V1H*, *ARTN2*, *ATP6V0A4*, *KIAA0195*, *MIR331* and *MIR4280*. Based on exposure-stratified analyses in the meta-analysis of the discovery cohorts, we further explored the effect sizes per exposure group. The lead SNPs that showed significant sleep $\times$ SNP interaction also showed effect estimates that modestly differed between LTST exposure groups (Supplementary Data 10). Interestingly, two lead SNPs near known HDL-c loci showed a 1df interaction $P$-value$_{FDR} < 0.05$, including SNPs near *CETP* and *LIPC* (Supplementary Data 4). Out of the seven previously unreported HDL-c loci identified in the joint model with STST, we found six loci with a 1df interaction $P$-value$_{FDR} < 0.05$, notably

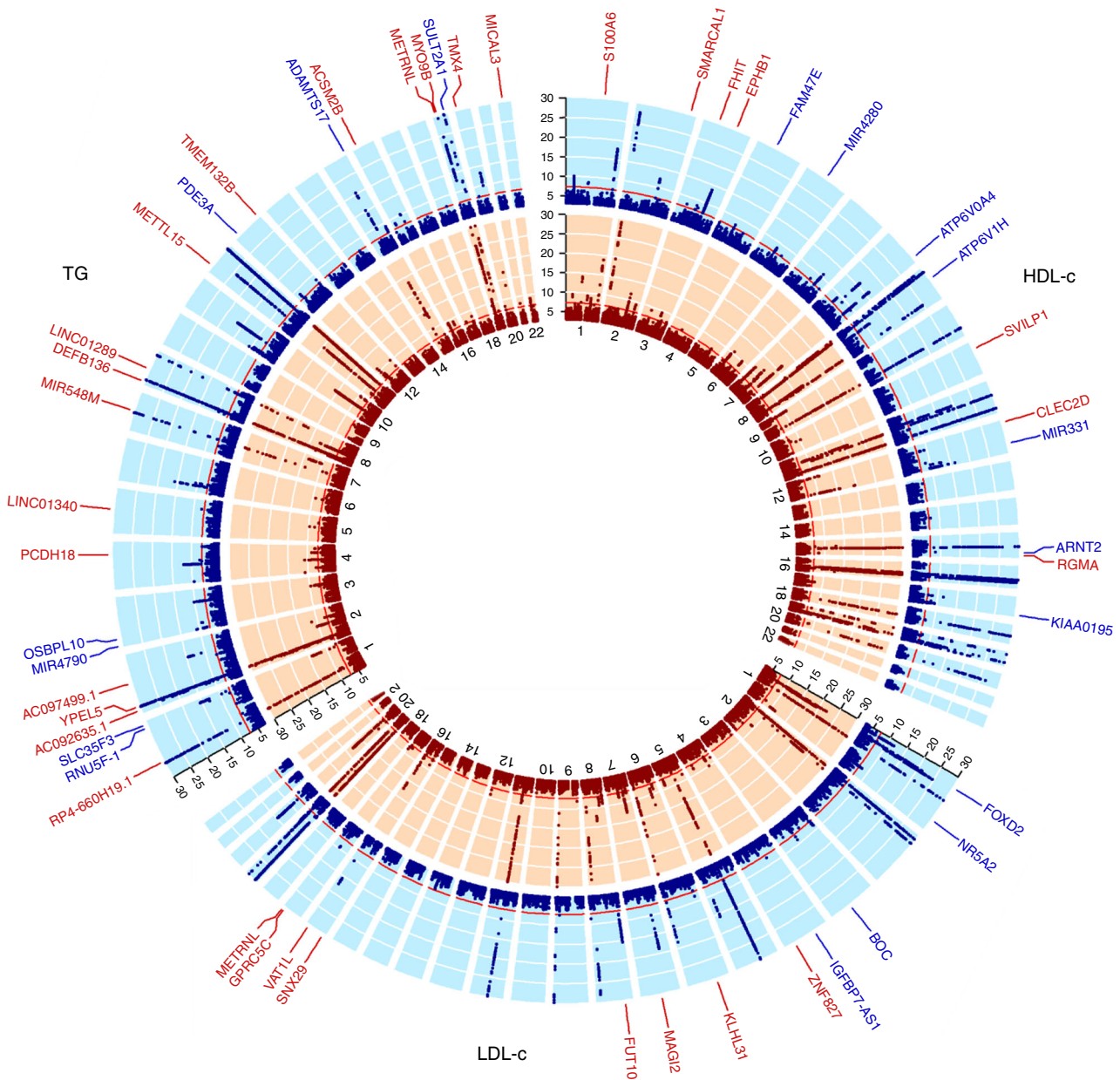

**Fig. 2** log(P-value of 2df interaction analyses) plots of the multi-ancestry analyses. Plot visualises the −log(P-values in the 2df interaction test) for HDL-c, LDL-c and TG per chromosome. In red (inner circle) are the −log(P-value) plots for the analyses taking into account potential interaction with short total sleep time. In blue (outer circle) are the −log(P-value plots for the analyses taking into account potential interaction with long total sleep time. Loci defined as novel and replicated are labelled. Replicated variants had to have 2df interaction test P-values of Stage $1 < 5 \times 10^{-7}$, Stage $2 < 0.05$ with a similar direction of effect as in the discovery meta-analysis and Stage $1 + 2 < 5 \times 10^{-8}$. Labelled gene names in red were identified in the STST analysis; labelled gene names in blue were identified in the LTST analysis. All −log(P-value in the 2df interaction test) > 30 were truncated to 30 for visualisation purposes only. The unlabelled regions with $P < 5 \times 10^{-8}$ in the 2df interaction test were in known loci. Figure prepared using the R package circlize[104]

lead SNPs mapped to *S1000A6*, *SMARCAL1*, *RGMA*, *EPHB1*, *FHIT* and *CLEC2D*. Again, their effect estimates differed between the exposure groups in the discovery multi-ancestry meta-analysis (Supplementary Data 11; Fig. 4). Some lead SNPs near known HDL-c loci showed evidence of a 1df interaction with STST (e.g., *MADD* and *LPL*; P-value$_{FDR}$ < 0.05).

For all four lead SNPs in previously unreported regions associated with LDL-c when considering LTST, we observed a 1df interaction P-value$_{FDR}$ < 0.05; notably, lead SNPs mapped to *IGFBP7-AS1*, *FOXD2*, *NR5A2* and *BOC*. One locus that mapped within a 1 Mb physical distance from known LDL-c locus (*PCSK9*) showed 1df interaction with LTST (Supplementary Data 4). Similarly, all eight lead SNPs in previously unreported

regions associated with LDL-c when considering STST, had a 1df interaction P-value$_{FDR}$ < 0.05; notably, lead SNPs mapped to *MAGI2*, *METRNL*, *VAT1L*, *FUT10*, *SNX29*, *ZNF827*, *GPRC5C* and *KLHL31*. In addition, of the known LDL-c loci, lead SNPs mapped within a physical distance of 1 Mb of *APOB* and *SLC22A1* showed a 1df interaction P-value$_{FDR}$ < 0.05 (Supplementary Data 5). For both analyses, we observed that effect estimates differed between the LTST and STST exposure groups in the multi-ancestry discovery analysis (Supplementary Data 10 and 11; Fig. 4).

All seven SNPs in previously unreported regions associated with TG when considering LTST, had a 1df interaction P-value$_{FDR}$ < 0.05; notably, lead SNPs mapped to *RNU5F*-1,

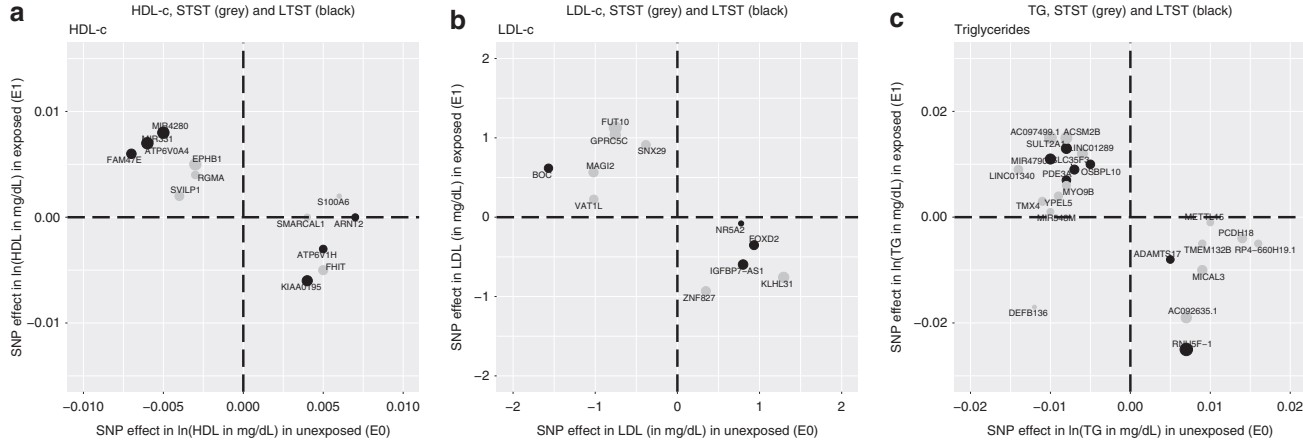

**Fig. 3** Sleep-interactions in known and previously unreported regions. Plot displaying the –log(P-value) of the 1df interaction between the SNP and either LTST or STST on the lipid trait after correction for multiple testing using false discovery rate against the allele frequency of the effect allele. Dotted horizontal line resembles the cut-off for the 1df interaction P-value$_{FDR}$ < 0.05 after correction for multiple testing using false discovery rate. In black are the novel loci for lipid traits; in grey are the identified lead SNPs mapped within a 1-Mb physical distance from a known lipid locus. Visualisation of the plots was performed using the R package ggplot2[105]. **a** HDL-c, long total sleep time; **b** HDL-c, short total sleep time; **c** LDL-c, long total sleep time; **d** LDL-c, short total sleep time; **e** triglycerides, long total sleep time; **f** triglycerides, short total sleep time

**Fig. 4** Comparison of SNP-main effects stratified by exposure. *X*-axis displays the effect sizes of the novel lead SNPs as observed in the meta-analyses of the unexposed individuals (LTST = '0', STST = '0'). *Y*-axis displays the effect sizes of the novel lead SNPs as observed in the meta-analyses of the exposed individuals (LTST = '1', STST = '1'). In black are the novel lead SNPs identified with LTST; in grey are the novel lead SNPs identified with STST. Sizes of the dots were weighted to the difference observed between exposed and unexposed. Visualisation of the plots was performed using the R package ggplot2[105]. **a** HDL-c; **b** LDL-c; **c** triglycerides

*SULT2A1*, *MIR4790*, *PDE3A*, *SLC35F3*, *ADAMTS17* and *OSBPL10*. In addition, we found some evidence for long sleep–SNP interaction in lead SNPs near known TG loci, including lead SNPs near *AKR1C4* and *NAT2* (Supplementary Data 4). Of the 16 lead SNPs in previously unreported regions associated with TG when considering STST, we observed 12 lead SNPs with a 1df interaction *P*-value $< 5 \times 10^{-4}$ (*P*-value$_{FDR}$ < 0.05), including lead SNPs mapped to *LINC0140*, *METRNL*, *AC092635.1*, *MICAL3*, *MIR548M*, *MYO9B*, *YPEL5*, *LINC01289*, *TMEM132B*, *ACSM2B*, *AC097499.1* and *RP4–660H19.1*. In addition, we observed some lead SNPs within 1 Mb physical distance from known TG loci, such as *MMP3* and *NECTIN2* (Supplementary Data 5). For both LTST and STST analyses, we again observed differing effects dependent on the exposure group in the discovery meta-analyses (Supplementary Data 10 and 11; Fig. 4).

**Look-ups and bioinformatics analyses.** Based on the lead SNPs mapped to the previously unreported loci, we conducted a look-up in GWAS summary statistics data on different questionnaire-based sleep phenotypes from up to 337,074 European-ancestry individuals of the UK Biobank (Supplementary Data 12). We only observed the TG-identified rs7924896 (*METTL15*) to be associated with snoring (*P*-value $= 1e^{-5}$) after correction for a total of 343 explored SNP–sleep associations (seven sleep phenotypes × 49 genes; ten SNPs were unavailable; threshold for significance = $1.46e^{-4}$). Furthermore, we did not observe that any of these identified SNPs was associated with accelerometer-based sleep traits (Supplementary Data 13). In general, we did not find substantial evidence that the identified lead SNPs in previously unreported regions were associated with coronary artery disease in the CARGIoGRAMplusC4D consortium (Supplementary Table 5).

Identified lipid loci from previously unreported regions were further explored in the GWAS catalogue (Supplementary Data 14). Several of the mapped genes of these lead SNPs have previously been identified with multiple other traits, such as body mass index (*FHIT*, *KLH31*, *ADAMTS17*, and *MAGI2*), mental health (*FHIT* [autism/schizophrenia, depression], *SNX13* [cognition]), gamma-glutamyltransferase (*ZNF827*, *MICAL3*), and inflammatory processes (*ZNF827*, *NR5A2*).

We additionally investigated differential expression of these lead SNPs using data from multiple tissues from the GTEx consortium[37,38] (Supplementary Data 15). Lead SNPs were frequently associated with mRNA expression levels of the mapped gene and with trans-eQTLs. For example, rs429921 (mapped to *VAT1L*) was associated with differential mRNA expression levels of *CLEC3A* and *WWOX*, which are located more upstream on chromosome 16 (Supplementary Fig. 6). rs3826692 (mapped to *MYO9B*) was specifically associated with differential expression of the nearby *USE1* gene. Identified SNPs were frequently associated with differential expression in the arteries. For example, rs6501801 (*KIAA0195*) was associated with differential expression in arteries at different locations. Several of the other identified SNPs showed differential expression in multiple tissues, including the gastrointestinal tract, (subcutaneous/visceral) adipose tissue, brain, heart, muscle, lung, liver, nervous system, skin, spleen, testis, thyroid and whole blood.

## Discussion

We investigated SNP–sleep interactions in a large, multi-ancestry, meta-analysis of blood lipid levels. Given the growing evidence that sleep influences metabolism[39–44], at least in part through effects on gene expression, we hypothesised that short/long habitual sleep duration may modify the effects of genetic loci on lipid levels. In a total study population of 126,926 individuals from five different ancestry groups, we identified 49 loci previously unreported in relation to lipid traits when considering either long or short total sleep time in the analyses. An additional ten previously unreported lipid loci were identified in analyses in Europeans only. Of these identified loci, most loci at least in part were driven by differing effects in short/long sleepers compared with the rest of the study population. Multiple of the genes identified from previously unreported regions for lipid traits have been previously identified in relation to adiposity, hepatic function, inflammation or psychosocial traits, collectively contributing to potential biological mechanisms involved in sleep-associated adverse lipid profile.

In addition to the over 300 genetic loci that already have been identified in relation to blood lipid concentrations in different efforts[4–10], we identified 49 additional loci associated with either HDL-c, LDL-c or TG in our multi-ancestry analysis. While for some of the SNPs had no neighbouring SNPs in high LD (e.g., rs7799249; mapped to *ATP6V0A4*), our applied filters (e.g., imputation quality > 0.5) would suggest that the chance of invalidity of the findings is negligible. Furthermore, in the case of rs7799249, no SNPs in high LD are known in individuals from different ancestries[45]. Considering the previously unreported TG loci identified by considering interactions with total sleep duration explain an additional 4.25% and 1.51% of the total variation in TG concentrations, for STST and LTST, respectively. Whilst the additionally explained variance for LDL-c (0.38% and 0.13%) and HDL-c (1.00% and 0.97%) was low/modest, the lead SNPs from previously unreported regions for LDL-c levels map to genes that are known to be associated with adiposity, inflammatory disorders, cognition, and liver function, thus identifying pathways by which sleep disturbances may influence lipid biology.

Across multiple populations, both short and long sleep duration have been associated with cardiovascular disease and diabetes[46]. There are numerous likely mechanisms for these associations. Experimental sleep loss results in inflammation, cellular stress in brain and peripheral tissues, and altered expression of genes associated with oxidative stress[47,48]. The impact of long sleep on metabolism is less well understood than the effect of short sleep, and multiple of the associations seem to overlap with short sleep as well. Long sleep duration is associated with decreased energy expenditure, increased sedentary time, depressed mood and obesity-related factors associated with inflammation and a pro-thrombotic state[49], as well as with higher C-reactive protein and interleukin-6 concentrations[50]. However, studies that adjusted for multiple confounders, including obesity, depression and physical activity, showed that long sleep remained a significant predictor of adverse cardiovascular outcomes[46,51]. Therefore, the adverse effects of long sleep also may partly reflect altered sleep–wake rhythms and chronodisruption resulting from misalignment between the internal biological clock with timing of sleep and other behaviours that track with sleep, such as timing of food intake, activity and light exposure[52]. Altered sleep–wake and circadian rhythms influence glucocorticoid signalling and autonomic nervous system excitation patterns across the day[41], which can influence the phase of gene expression. These inputs appear to be particularly relevant for genes controlling lipid biosynthesis, absorption and degradation, many of which are rhythmically regulated and under circadian control[53]. Moreover, the molecular circadian clock acts as a rate-limiting step in cholesterol and bile synthesis, supporting the potential importance of circadian disruption in lipid biology[54]. Collectively, these data suggest different biological mechanisms involved in short and long sleep-associated adverse lipid profiles.

Consistent with different hypothesised physiological effects of short and long sleep, we observed no overlap in the previously

unreported loci that were identified by modelling interactions with short or long sleep duration. The lipid loci that were identified after considering STST include *FHIT*, *MAGI2* and *KLH3*, which have been previously associated with body mass index (BMI)[55–61]. Interestingly, although not genome-wide significant, variation in *MAGI2* has been associated with sleep duration[62], however, we did not find evidence for an association with rs10244093 in *MAGI2* with any sleep phenotype in the UK Biobank sample. Variants in *MICAL3* and *ZNF827*, that were also identified after considering STST, have been associated with serum liver enzymes gamma-glutamyltransferase measurement and/or aspartate aminotransferase levels[63,64], which have been implicated in cardiometabolic disturbances[65–68] and associated with prolonged work hours (which often results in short or irregular sleep)[69]. Other loci identified through interactions with STST were in genes previously associated with neurocognitive and neuropsychiatric conditions, possibly reflecting associations mediated by heightened levels of cortisol and sympathetic activity that frequently accompany short sleep.

In relation to LTST, the previously unreported lipid genes have been previously related to inflammation-driven diseases of the intestine, blood pressure and blood count measurements, including traits influenced by circadian rhythms[70,71]. However, none of these loci with LTST directly interacted with genes involved in the central circadian clock (e.g., *PER2*, *CRY2* and *CLOCK*) in the KEGG pathways database[72]. The *NR5A2* and *SLC35F3* loci have been associated with inflammation-driven diseases of the intestine[73,74]. Ulcerative colitis, an inflammatory bowel disease, has been associated with both longer sleep duration[75] and circadian disruption[70]. *ARNT2*, also identified via a LTST interaction, heterodimerizes with transcriptional factors implicated in homoeostasis and environmental stress responses[76,77]. A linkage association study has reported nominal association of this gene with lipids in a Caribbean Hispanic population[78].

We identified a number of additional genetic lead SNPs in the meta-analyses performed in European-Americans only. For example, we identified rs3938236 mapped to *SPRED1* to be associated with HDL-c after accounting for potential interaction with LTST. Interestingly, this gene has been previously associated with hypersomnia in Caucasian and Japanese populations[79], but was not identified in our larger multi-ancestry analysis, possibly due to cultural differences in sleep behaviours[80].

We additionally found evidence, amongst others, in the known lipid loci *APOB*, *PCSK9* and *LPL* for interaction with either short or long sleep. Associations have been observed previously between short sleep and ApoB concentrations, have been observed previously[81]. LPL expression has been shown to follows a diurnal rhythm in several metabolic organs[43,82], and disturbing sleeping pattern by altered light exposure can lower LPL activity, at least in brown adipose tissue[43]. Similar effects of sleep on hepatic secretion of ApoB and PCSK9 may be expected. Indeed, in humans PCSK9 has a diurnal rhythm synchronous with hepatic cholesterol synthesis[83]. Although the interaction effects we observed were rather weak, the supporting evidence from the literature suggests that sleep potentially modifies the effect of some of the well-known lipid regulators that are also targets for therapeutic interventions.

Some of the previously unreported lipid loci have been previously associated with traits related to sleep. For example, *MAGI2* and *MYO9B*[62] have been suggestively associated with sleep duration and quality, respectively. Genetic variation in *TMEM132B* has been associated with excessive daytime sleepiness[84], and *EPHB1* has been associated with self-reported chronotype[85]. These findings suggest some shared genetic component of lipid regulation and sleep biology. However, with the exception of the *METTL15*-mapped rs7924896 variant in relation to snoring, none of the lead SNPs mapped to the previously unreported lipid loci were associated with any of the investigated sleep phenotypes in the UK Biobank population, suggesting no or minimal shared component in sleep and lipid biology but rather that sleep duration specifically modifies the effect of the variant on the lipid traits.

This study was predominantly comprised of individuals of European ancestry, despite our efforts to include as many studies of diverse ancestries as possible. For this reason, additional efforts are required to specifically study gene–sleep interactions in those of African, Asian and Hispanic ancestry once more data becomes available. In line, we identified several loci that were identified only in the European-ancestry analysis, and not in the multi-ancestry analysis, suggestion that there might be ancestry-specific effects. The multi-ancestry analysis highlighted the genetic regions that are more likely to play a role in sleep-associated adverse lipid profiles across ancestries. In addition, our study used questionnaire-based data on sleep duration. Although the use of questionnaires likely increased measurement error and decreased statistical power, questionnaire-based assessments of sleep duration have provided important epidemiological data, including the identification of genetic variants for sleep traits in genome-wide association studies[84]. Identified variants for sleep traits have been recently successfully validated using accelerometer data[86], although the overall genetic correlation with accelerometer-based sleep duration was shown to be low[87]. Moreover, observational studies showed only a modest correlation between the phenotypes[88], which suggest that each approach characterises somewhat different phenotypes. At this time, we did not have sufficient data to evaluate other measures of sleep duration such as polysomnography or accelerometery. A more comprehensive characterisation, additional circadian traits as well as larger study samples (e.g., embedded in the large biobanks that become increasingly available for research) will refine our understanding of the interaction of these fundamental phenotypes and lipid biology.

In summary, the gene–sleep interaction efforts described in the present multi-ancestry study identified many lipid loci previously unreported to be associated with either HDL-c, LDL-c or triglycerides levels. Multiple of the these loci were driven by interactions with either short or long sleep duration, and were mapped to genes also associated with adiposity, inflammatory or neuropsychiatric traits. Collectively, the results highlight the interactions between extreme sleep–wake exposures and lipid biology.

## Methods

**Participants**. Analyses were performed locally by the different participating studies. Discovery and replication analyses comprised men and women between the age of 18 and 80 years, and were conducted separately for the different contributing (self-defined) ancestry groups, including: European, African, Asian, Hispanic and Brazilian (discovery analysis only). Descriptions of the different participating studies are described in detail in the Supplementary Notes 1 and 3, and study-specific characteristics (sizes, trait distribution and data preparation) are presented in Supplementary Tables 1–6. Every effort was made to include as many studies as possible.

**Ethical regulations**. The present work was approved by the Institutional Review Board of Washington University in St. Louis and complies with all relevant ethical regulations. Each participating study obtained written informed consent from all participants and received approval from the appropriate local institutional review boards.

**Lipid traits**. We conducted all analyses on the following lipid traits: HDL-c, LDL-c and TG. TG and LDL-c concentrations were measured in samples from individuals who had fasted for at least 8 hours. LDL-c could be either directly assayed or derived using the Friedewald equation[89] (the latter being restricted to those with TG ≤ 400 mg/dL). We furthermore corrected LDL-c for the use of lipid-lowering drugs, defined as any use of a statin drug or any unspecified lipid-lowering drug

after the year 1994 (when statin use became common in general practice). If LDL-c was directly assayed, the concentration of LDL-c was corrected by dividing the LDL-c concentration by 0.7. If LDL-c was derived using the Friedewald equation, we first divided the concentration of total cholesterol by 0.8 before LDL-c was calculated by the Friedewald equation. Due to the skewed distribution of HDL-c and TG, we ln-transformed the concentration prior to the analyses; no transformation for LDL-c was required. When an individual cohort measured the lipid traits during multiple visits, the visit with the largest available sample and concurrent availability of the sleep questions was selected.

**Nocturnal total sleep time.** Contributing cohorts collected information on the habitual sleep duration using either a single question such as 'on an average night, how long do you sleep?' or as part of a standardised sleep questionnaire (e.g., the Pittsburgh Sleep Quality Index questionnaire[90]). For the present project, we defined both STST and LTST. To harmonise the sleep duration data across cohorts from different countries, cultures and participants with different physical characteristics, in whom sleep duration was assessed using various questions, we defined STST and LTST using cohort-specific residuals, adjusting for age and sex. An exception was for AGES and HANDLS cohorts, we used a cohort-specific definition due to limited response categories in relationship to the available question on sleep duration. Instead, we defined STST or LTST based on expert input. Exposure to STST was defined as the lowest 20% of the sex- and age-adjusted sleep-time residuals (coded as '1'). Exposure to LTST was defined as the highest 20% of the sex- and age-adjusted sleep-time residuals (coded as '1'). For both sleep-time definitions, we considered the remaining 80% of the population as being unexposed to either STST or LTST (coded as '0').

**Genotype data.** Genotyping was performed by each participating study locally using genotyping arrays from either Illumina (San Diego, CA, USA) or Affymetrix (Santa Clara, CA, USA). Each study conducted imputation using various software programmes and with local cleaning thresholds for call rates (usually > 98%) and Hardy–Weinberg equilibrium (usually $P$-value < 1e$^{-5}$). The cosmopolitan reference panel from the 1000 Genomes Project Phase I Integrated Release Version 3 Haplotypes (2010–11 data freeze, 2012-03-14 haplotypes) was specified for imputation. Only SNPs on the autosomal chromosomes with a minor allele frequency of at least 0.01 were considered in the analyses. Specific details of each participating study's genotyping platform and imputation software are described (Supplementary Tables 3 and 6).

**Stage 1 analysis (discovery phase).** The discovery phase of the present project included 21 cohorts contributing data from 28 study/ancestry groups, and included up to 62,457 participants of EUR, AFR, ASN, HISP and BR ancestry (Supplementary Tables 1–3). All cohorts ran statistical models according to a standardised analysis protocol. The main model for this project examined the SNP-main effect and the multiplicative interaction term between the SNP and either LTST or STST:

$$E(Y) = \beta_0 + \beta_E E + \beta_G SNP + \beta_{GE} E * SNP + \beta_C C \qquad (1)$$

in which E is the sleep exposure variable (LTST/STST) and C are the (study-specific) covariates, which was similar to what we have done in previous studies[4,11,12]. In addition, we examined the SNP-main effect (without incorporating LTST/STST) and the SNP-main effect stratified by the exposure:

$$E(Y) = \beta_{0*} + \beta_{G*} SNP + \beta_{C*} C \qquad (1)$$

All models were performed for each lipid trait and separately for the different ancestry groups. Consequently, per ancestry group, we requested a total of seven GWA analyses per lipid trait. All models were adjusted for age, sex, field centre (if required), and the first principal components to correct for population stratification. The number of principal components included in the model was chosen according to cohort-specific preferences (ranging from 0 to 10). All studies were asked to provide the effect estimates (SNP-main and -interaction effect) with accompanying robust estimates of the standard error for all requested models. A robust estimate of the covariance between the main and interaction effects was also provided. To obtain robust estimates of covariance matrices and standard errors, studies with unrelated participants used R packages such as either sandwich[91,92] or ProbABEL[93]. Studies including related individuals used either generalised estimating equations (R package geepack[94]) or linear mixed models (GenABEL[95], MMAP or R package sandwich[91,92]). Sample code provided to studies to generate these data has been previously published[96].

Upon completion of the analyses by local institution, all summary data were stored centrally for further processing and meta-analyses. We performed estimative quality control (QC) using the R-based package EasyQC[97] (www.genepi-regensburg.de/easyqc) at the study level (examining the results of each study individually), and subsequently at the ancestry level (after combining all ancestry-specific cohorts using meta-analyses). Study-level QC consisted of excluding all SNPs with MAF < 0.01, harmonisation of alleles, comparison of allele frequencies with ancestry-appropriate 1000 Genomes reference data, and harmonisation of all SNPids to a standardised nomenclature according to chromosome and position. Ancestry-level QC included the compilation of summary statistics on all effect estimates, standard errors and p-values across studies to identify potential outliers,

and production of SE-N and QQ plots to identify analytical problems (such as improper trait transformations)[98].

Prior to the ancestry-specific meta-analyses, we excluded the following SNPs from the cohort-level data files: all SNPs with an imputation quality < 0.5, and all SNPs with a minor allele count in the exposed group (LTST or STST equals '1') x imputation quality of less than 20. SNPs in the European-ancestry and multi-ancestry analyses had to be present in at least three cohorts and 5000 participants. Due to the limited sample size of the non-European ancestries (either discovery or replication), we did not take into account this filter in those ancestry-level meta-analyses.

Meta-analyses were conducted for all models using the inverse variance-weighted fixed effects method as implemented in METAL[99] (http://genome.sph.umich.edu/wiki/METAL). We evaluated both a 1df of freedom test of interaction effect and a 2df joint test of main and interaction effects, following previously published methods[29]. A 1df Wald test was used to evaluate the 1df interaction, as well as the main effect in models without an interaction term. A 2df Chi-squared test was used to jointly test the effects of both the variant and the variant × LTST/STST interaction[100]. Meta-analyses were conducted within each ancestry separately. Multi-ancestry meta-analyses were conducted on all ancestry-specific meta-analyses. Genomic control correction was applied on all cohorts incorporated in the ancestry-level meta-analyses as well as on the final meta-analyses for the publication. From this effort, we selected all SNPs associated with any of the lipid traits with $P \le 5 \times 10^{-7}$ in the 2df interaction test for replication in the Stage 2 analysis. This cut-off was selected to minimise false-negative results.

**Stage 2 analysis (replication phase).** All SNPs selected in Stage 1 for replication were evaluated in the interaction model in up to 18 cohorts contributing data from 20 study groups totalling up to 64,469 individuals (Supplementary Tables 4–6). As we had a limited number of individuals from non-European ancestry in the replication analyses, we did not consider an the non-European ancestries separately and only focussed on a European-ancestry and multi-ancestry analysis.

Study- and ancestry-level QC was carried out as in stage 1. In contrast to stage 1, no additional filters were included for the number of studies or individuals contributing data to stage 2 meta-analyses, as these filters were implemented to reduce the probability of false positives, and were less relevant in stage 2. Stage 2 SNPs were evaluated in all ancestry groups and for all traits, no matter what specific meta-analysis met the $P$-value threshold in the stage 1 analysis. We did not apply genomic control to any of the Stage 2 analyses given the expectation of association.

An additional meta-analysis was performed combining the Stage 1 and 2 meta-analyses. SNPs (irrespective of being known or previously unreported) were considered to be replicated when the 2df interaction test $P$-values of Stage 1 < 5 × 10$^{-7}$, Stage 2 < 0.05 with a similar direction of effect as in the discovery meta-analysis, and Stage 1 + 2 < 5 × 10$^{-8}$. Replicated SNPs were subsequently used in different bioinformatics tools for further processing. In addition, 1df $P$-values (SNP-sleep interaction effect only) of the lead SNPs of both the replicated known and previously unreported loci were calculated to explore whether genetic variant were specifically driven by SNP-main or SNP-interaction effects. Based on the total number of lead SNPs across all analyses, we performed correction using the false discovery rate to quantify statistical significance[36].

**Bioinformatics.** Replicated SNPs were first processed using the online tool FUMA[101] to identify independent lead SNPs and to perform gene mapping. From the SNP that has a $P$-value in the 2df interaction test < 5 × 10$^{-8}$, we determined lead SNPs that were independent from each other at $R^2$ < 0.1 using the 1000 G Phase 3 EUR as a reference panel population. Independent lead SNPs with a physical distance > 1 mB from a known locus were considered as previously unreported. Regional plots of these loci were made using the online LocusZoom tool[102]. The explained variance of the identified genetic lead SNPs mapped to previously unreported lipid regions was calculated based on the summary statistics of the combined analysis of Stage 1 and 2 using the R-based VarExp package, which has been previously validated to provide similar results to individual participant data[35]. This package calculates the variance explained on the basis of the combined (joint) SNP-main and SNP-interaction effect. Differential expression analyses of the lead SNPs in the identified genetic loci was performed using GTEx [https://gtexportal.org/home/][37,38].

**Look-ups of previously unreported loci in other databases.** The genetic loci for the three lipid traits previously unreported were further explored in the GWAS catalogue [https://www.ebi.ac.uk/gwas/] to investigate the role of these mapped genes in other traits. Furthermore, we extracted the lead SNPs from the previously unreported lipid loci from publically available GWAS data from the UK Biobank [http://www.nealelab.is/uk-biobank/] for different questionnaire-based sleep phenotypes, notably 'daytime snoozing/sleeping (narcolepsy)', 'getting up in the morning', 'morning/evening person (chronotype)', 'nap during the day', 'sleep duration', 'sleeplessness/insomnia' and 'snoring'. Analyses on these phenotypes were generally done using continuous outcomes; the variable 'sleep duration' was expressed in hours of total sleep per day. GWAS in the UK Biobank were done in European-ancestry individuals only (N up to 337,074). We furthermore extracted

the identified lead SNPs from the previously unreported regions for lipid traits from the GWAS analyses done on accelerometer-based sleep variables, which was done in European-ancestry individuals from the UK Biobank ($N = 85,670$; [http://sleepdisordergenetics.org/])[87]. In addition, we extracted the these identified lead SNPs from publically available summary-statistics data on coronary artery disease of the CARDIoGRAMplusC4D consortium, which included 60,801 cases of coronary artery disease and 123,504 controls [http://www.cardiogramplusc4d.org][103].

**Reporting summary**. Further information on research design is available in the Nature Research Reporting Summary linked to this article.

## Data availability

Due to restrictions in the written informed consent and local regulations, no individual genotype-level data could be shared that were part of this project. Summary results files from both the trans-ancestry and European meta-analyses are available to the public via the CHARGE (Cohorts for Heart and Ageing Research in Genomics Epidemiology) dbGaP summary site (phs000930 [https://www.ncbi.nlm.nih.gov/projects/gap/cgi-bin/study.cgi?study_id=phs000930.v1.p1]). We acknowledge the use of publically available data sources for summary-based statistics, which includes the gTex portal [https://gtexportal.org/home/], Nealelab [http://www.nealelab.is/uk-biobank/], Sleep Disorder Genetics [http://sleepdisordergenetics.org/] and the CARDIoGRAMplusC4D consortium [http://www.cardiogramplusc4d.org].

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

## Acknowledgements

This project was supported by a grant from the US National Heart, Lung, and Blood Institute (NHLBI) of the National Institutes of Health (R01HL118305). This research was supported in part by the Intramural Research Program of the National Human Genome Research Institute, National Institutes of Health. Tuomas O. Kilpeläinen was supported by the Danish Council for Independent Research (DFF–6110-00183) and the Novo Nordisk Foundation (NNF18CC0034900, NNF17OC0026848 and NNF15CC0018486). Diana van Heemst was supported by the European Commission funded project HUMAN (Health-2013-INNOVATION-1-602757). Susan Redline was supported in part by NIH R35HL135818 and HL11338. Study-specific acknowledgements can be found in the Supplementary Notes 2 and 4. The data on coronary artery disease have been contributed by the Myocardial Infarction Genetics and CARDIoGRAM investigators, and have been downloaded from www.CARDIOGRAMPLUSC4D.ORG.

## Author contributions

R.N. and M.M.B. conducted the centralised data analysis, which included quality control checks, meta-analyses and bioinformatics. R.N., M.M.B. and S.R. drafted the initial version of the paper. R.N., M.M.B., H.W., T.W.W., A.R.B., T.O.K., P.B.M., C.T.L., A.C. M., D.C.R., D.v.H. and S.R. were part of the writing group and were mainly responsible for the study design, interpretation of the data and critical commenting on the initial draft versions of the paper. All other co-authors were responsible for cohort-level data collection, cohort-level data analysis and critical reviews of the draft paper. All authors approved the final version of the paper that was submitted to the journal.

## Competing interests

D.O.M.K. is a part-time research consultant for Metabolon, Inc. H.J.G. has received travel grants and speakers honoraria from Fresenius Medical Care, Neuraxpharm and Janssen Cilag. H.J.G. has received research funding from the German Research Foundation (DFG), the German Ministry of Education and Research (BMBF), the DAMP Foundation, Fresenius Medical Care, the EU "Joint Programme Neurodegenerative Disorders (JPND)" and the European Social Fund (ESF)". S.A. reports employment and stock options with 23andMe, Inc.

## Additional information

Raymond Noordam[1,116]*, Maxime M. Bos[1,116], Heming Wang[2,116], Thomas W. Winkler[3,116], Amy R. Bentley[4,116], Tuomas O. Kilpeläinen[5,6,116], Paul S. de Vries[7], Yun Ju Sung[8], Karen Schwander[8], Brian E. Cade[2], Alisa Manning[9,10], Hugues Aschard[11,12], Michael R. Brown[7], Han Chen[7,13], Nora Franceschini[14], Solomon K. Musani[15], Melissa Richard[16], Dina Vojinovic[17], Stella Aslibekyan[18], Traci M. Bartz[19], Lisa de las Fuentes[8,20], Mary Feitosa[21], Andrea R. Horimoto[22], Marjan Ilkov[23], Minjung Kho[24], Aldi Kraja[21], Changwei Li[25], Elise Lim[26], Yongmei Liu[27], Dennis O. Mook-Kanamori[28,29], Tuomo Rankinen[30], Salman M. Tajuddin[31], Ashley van der Spek[17], Zhe Wang[7], Jonathan Marten[32], Vincent Laville[12], Maris Alver[33,34], Evangelos Evangelou[35,36], Maria E. Graff[14], Meian He[37], Brigitte Kühnel[38,39], Leo-Pekka Lyytikäinen[40,41], Pedro Marques-Vidal[42], Ilja M. Nolte[43], Nicholette D. Palmer[44], Rainer Rauramaa[45], Xiao-Ou Shu[46], Harold Snieder[43], Stefan Weiss[47], Wanqing Wen[46], Lisa R. Yanek[48], Correa Adolfo[15], Christie Ballantyne[49,50], Larry Bielak[24], Nienke R. Biermasz[51,52], Eric Boerwinkle[7,53], Niki Dimou[36], Gudny Eiriksdottir[23], Chuan Gao[54], Sina A. Gharib[55], Daniel J. Gottlieb[2,10,56], José Haba-Rubio[57], Tamara B. Harris[58], Sami Heikkinen[59,60], Raphaël Heinzer[57], James E. Hixson[7], Georg Homuth[47], M. Arfan Ikram[17,61], Pirjo Komulainen[45], Jose E. Krieger[22], Jiwon Lee[2], Jingmin Liu[62], Kurt K. Lohman[63], Annemarie I. Luik[17], Reedik Mägi[33], Lisa W. Martin[64], Thomas Meitinger[65], Andres Metspalu[33,34], Yuri Milaneschi[64], Mike A. Nalls[66,67], Jeff O'Connell[68,69], Annette Peters[39,70], Patricia Peyser[24], Olli T. Raitakari[71,72], Alex P. Reiner[62], Patrick C.N. Rensen[51,52], Treva K. Rice[8], Stephen S. Rich[73], Till Roenneberg[74], Jerome I. Rotter[75], Pamela J. Schreiner[76], James Shikany[77], Stephen S. Sidney[78], Mario Sims[15], Colleen M. Sitlani[79], Tamar Sofer[2,80], Konstantin Strauch[81,82], Morris A. Swertz[83], Kent D. Taylor[75],

André G. Uitterlinden [17,84], Cornelia M. van Duijn [17,85], Henry Völzke [86], Melanie Waldenberger [38,39,70], Robert B. Wallance [87], Ko Willems van Dijk [51,52,88], Caizheng Yu [37], Alan B. Zonderman [89], Diane M. Becker [48], Paul Elliott [34,90,91,92], Tõnu Esko [33,93], Christian Gieger [38,94], Hans J. Grabe [95], Timo A. Lakka [45,60,96], Terho Lehtimäki [40,41], Kari E. North [14], Brenda W.J.H. Penninx [97], Peter Vollenweider [42], Lynne E. Wagenknecht [98], Tangchun Wu [37], Yong-Bing Xiang [99], Wei Zheng [46], Donna K. Arnett [100], Claude Bouchard [30], Michele K. Evans [31], Vilmundur Gudnason [23,101], Sharon Kardia [24], Tanika N. Kelly [102], Stephen B. Kritchevsky [103], Ruth J.F. Loos [104,105], Alexandre C. Pereira [22], Mike Province [21], Bruce M. Psaty [106,107], Charles Rotimi [4], Xiaofeng Zhu [108], Najaf Amin [17], L. Adrienne Cupples [26,109], Myriam Fornage [16], Ervin F. Fox [110], Xiuqing Guo [75], W. James Gauderman [111], Kenneth Rice [112], Charles Kooperberg [62], Patricia B. Munroe [113,114], Ching-Ti Liu [26], Alanna C. Morrison [7], Dabeeru C. Rao [8], Diana van Heemst [1] & Susan Redline [2,115]*

[1]Department of Internal Medicine, Section of Gerontology and Geriatrics, Leiden University Medical Center, Leiden, The Netherlands. [2]Division of Sleep and Circadian Disorders, Harvard Medical School, Brigham and Women's Hospital, Boston, MA, USA. [3]Department of Genetic Epidemiology, University of Regensburg, Regensburg, Germany. [4]Center for Research on Genomics and Global Health, National Human Genome Research Institute, National Institutes of Health, Bethesda, MD, USA. [5]Novo Nordisk Foundation Center for Basic Metabolic Research, Faculty of Health and Medical Sciences, University of Copenhagen, Copenhagen 2200, Denmark. [6]Department of Environmental Medicine and Public Health, The Icahn School of Medicine at Mount Sinai, New York, New York, USA. [7]Human Genetics Center, Department of Epidemiology, Human Genetics, and Environmental Sciences, School of Public Health, The University of Texas Health Science Center at Houston, Houston, TX, USA. [8]Division of Biostatistics, Washington University School of Medicine, St. Louis, MO, USA. [9]Clinical and Translational Epidemiology Unit, Massachusetts General Hospital, Boston, MA, USA. [10]Department of Medicine, Harvard Medical School, Boston, MA, USA. [11]Department of Epidemiology, Harvard School of Public Health, Boston, MA, USA. [12]Centre de Bioinformatique, Biostatistique et Biologie Intégrative (C3BI), Institut Pasteur, Paris, France. [13]Center for Precision Health, School of Public Health & School of Biomedical Informatics, The University of Texas Health Science Center at Houston, Houston, TX, USA. [14]Department of Epidemiology, Gillings School of Global Public Health, University of North Carolina, Chapel Hill, NC, USA. [15]Jackson Heart Study, Department of Medicine, University of Mississippi Medical Center, Jackson, MS, USA. [16]Brown Foundation Institute of Molecular Medicine, The University of Texas Health Science Center at Houston, Houston, TX, USA. [17]Department of Epidemiology, Erasmus University Medical Center, Rotterdam, The Netherlands. [18]Department of Epidemiology, University of Alabama at Birmingham, Birmingham, AL, USA. [19]Cardiovascular Health Research Unit, Biostatistics and Medicine, University of Washington, Seattle, WA, USA. [20]Cardiovascular Division, Department of Medicine, Washington University School of Medicine, St. Louis, MO, USA. [21]Division of Statistical Genomics, Department of Genetics, Washington University School of Medicine, St. Louis, MO, USA. [22]Laboratory of Genetics and Molecular Cardiology, Heart Institute (InCor), University of São Paulo Medical School, São Paulo, SP, Brazil. [23]Icelandic Heart Association, Kopavogur, Iceland. [24]Department of Epidemiology, School of Public Health, University of Michigan, Ann Arbor, MI, USA. [25]Epidemiology and Biostatistics, University of Georgia at Athens College of Public Health, Athens, GA, USA. [26]Department of Biostatistics, Boston University School of Public Health, Boston, MA, USA. [27]Public Health Sciences, Epidemiology and Prevention, Wake Forest University Health Sciences, Winston-Salem, NC, USA. [28]Department of Clinical Epidemiology, Leiden University Medical Center, Leiden, Netherlands. [29]Department of Public Health and Primary Care, Leiden University Medical Center, Leiden, Netherlands. [30]Human Genomics Laboratory, Pennington Biomedical Research Center, Baton Rouge, LA, USA. [31]Health Disparities Research Section, Laboratory of Epidemiology and Population Sciences, National Institute on Aging, National Institutes of Health, Baltimore, MD, USA. [32]Medical Research Council Human Genetics Unit, Institute of Genetics and Molecular Medicine, Institute of Genetics and Molecular Medicine, University of Edinburgh, Edinburgh, UK. [33]Estonian Genome Center, Institute of Genomics, University of Tartu, Tartu, Estonia. [34]Department of Biotechnology, Institute of Molecular and Cell Biology, University of Tartu, Tartu, Estonia. [35]Department of Epidemiology and Biostatistics, School of Public Health, Imperial College London, London, UK. [36]Department of Hygiene and Epidemiology, University of Ioannina Medical School, Ioannina, Greece. [37]Department of Occupational and Environmental Health and State Key Laboratory of Environmental Health for Incubating, Tongji Medical College, Huazhong University of Science and Technology, Wuhan, China. [38]Research Unit of Molecular Epidemiology, Helmholtz Zentrum München, German Research Center for Environmental Health, Neuherberg, Germany. [39]Institute of Epidemiology, Helmholtz Zentrum München, German Research Center for Environmental Health, Neuherberg, Germany. [40]Department of Clinical Chemistry, Fimlab Laboratories, Tampere, Finland. [41]Department of Clinical Chemistry, Finnish Cardiovascular Research Center—Tampere, Faculty of Medicine and Health Technology, Tampere University, Tampere, Finland. [42]Medicine, Internal Medicine, Lausanne University Hospital, Lausanne, Switzerland. [43]University of Groningen, University Medical Center Groningen, Department of Epidemiology, Groningen, The Netherlands. [44]Biochemistry, Wake Forest School of Medicine, Winston-Salem, NC, USA. [45]Foundation for Research in Health Exercise and Nutrition, Kuopio Research Institute of Exercise Medicine, Kuopio, Finland. [46]Division of Epidemiology, Department of Medicine, Vanderbilt University School of Medicine, Nashville, TN, USA. [47]Interfaculty Institute for Genetics and Functional Genomics, Department of Functional Genomics, University Medicine Greifswald, Greifswald, Germany. [48]Division of General Internal Medicine, Department of Medicine, Johns Hopkins University School of Medicine, Baltimore, MD, USA. [49]Section of Cardiovascular Research, Baylor College of Medicine, Houston, TX, USA. [50]Houston Methodist Debakey Heart and Vascular Center, Houston, TX, USA. [51]Department of Internal Medicine, Division of Endocrinology, Leiden University Medical Center, Leiden, The Netherlands. [52]Einthoven Laboratory for Experimental Vascular Medicine, Leiden, The Netherlands. [53]Human Genome Sequencing Center, Baylor College of Medicine, Houston, TX, USA. [54]Molecular Genetics and Genomics Program, Wake Forest School of Medicine, Winston-Salem, NC, USA. [55]Computational Medicine Core, Center for Lung Biology, UW Medicine Sleep Center, Medicine, University of Washington, Seattle, WA, USA. [56]VA Boston Healthcare System, Boston, MA, USA. [57]Medicine, Sleep Laboratory, Lausanne University Hospital, Lausanne, Switzerland. [58]Laboratory of Epidemiology and Population Sciences, National Institute on Aging, National Institutes of Health, Bethesda, MD, USA. [59]Institute of Clinical Medicine, Internal Medicine, University of Eastern Finland, Kuopio, Finland. [60]Institute of Biomedicine, School of Medicine, University of Eastern Finland, Kuopio Campus, Finland. [61]Department of Radiology and Nuclear Medicine, Erasmus University Medical Center, Rotterdam, The Netherlands. [62]Fred Hutchinson Cancer Research Center, University of Washington School of Public Health, Seattle, WA, USA. [63]Public Health Sciences, Biostatistical Sciences, Wake Forest University Health Sciences, Winston-Salem, NC, USA. [64]Cardiology, School of Medicine and Health Sciences, George Washington University, Washington, D.C., USA. [65]Institute of Human Genetics, Helmholtz Zentrum München, German Research

Center for Environmental Health, Neuherberg, Germany. [66]Laboratory of Neurogenetics, National Institute on Aging, Bethesda, MD, USA. [67]Data Tecnica International, Glen Echo, MD, USA. [68]Division of Endocrinology, Diabetes, and Nutrition, University of Maryland School of Medicine, Baltimore, MD, USA. [69]Program for Personalized and Genomic Medicine, University of Maryland School of Medicine, Baltimore, MD, USA. [70]DZHK (German Centre for Cardiovascular Research), partner site Munich Heart Alliance, Neuherberg, Germany. [71]Department of Clinical Physiology and Nuclear Medicine, Turku University Hospital, Turku, Finland. [72]University of Turku, Turku, Finland. [73]Center for Public Health Genomics, University of Virginia, Charlottesville, VA, USA. [74]Institute of Medical Psychology, Ludwig-Maximilians-Universitat Munchen, Munich, Germany. [75]Genomic Outcomes, Department of Pediatrics, Institute for Translational Genomics and Population Sciences, LABioMed at Harbor-UCLA Medical Center, Torrance, CC, USA. [76]Division of Epidemiology & Community Health, School of Public Health, University of Minnesota, Minneapolis, MN, USA. [77]Division of Preventive Medicine, Department of Medicine, University of Alabama at Birmingham, Birmingham, AL, USA. [78]Division of Research, Kaiser Permanente Northern California, Oakland, CA, USA. [79]Cardiovascular Health Research Unit, Medicine, University of Washington, Seattle, WA, USA. [80]Institute of Human Genetics, Technische Universität München, Munich, Germany. [81]Institute of Genetic Epidemiology, Helmholtz Zentrum München, German Research Center for Environmental Health, Neuherberg, Germany. [82]Institute for Medical Informatics Biometry and Epidemiology, Ludwig-Maximilians-Universitat Munchen, Munich, Germany. [83]University of Groningen, University Medical Center Groningen, Department of Genetics, Groningen, The Netherlands. [84]Department of Internal Medicine, Erasmus University Medical Center, Rotterdam, The Netherlands. [85]Nuffield Department of Population Health, University of Oxford, Oxford, UK. [86]Institute for Community Medicine, University Medicine Greifswald, Greifswald, Germany. [87]Department of Epidemiology, University of Iowa College of Public Health, Iowa City, IA, USA. [88]Department of Human Genetics, Leiden University Medical Center, Leiden, The Netherlands. [89]Behavioral Epidemiology Section, Laboratory of Epidemiology and Population Sciences, National Institute on Aging, National Institutes of Health, Baltimore, MD, USA. [90]MRC-PHE Centre for Environment and Health, School of Public Health, Imperial College London, London, UK. [91]National Institute of Health Research Imperial College London Biomedical Research Centre, London, UK. [92]UK-DRI Dementia Research Institute at Imperial College London, London, UK. [93]Broad Institute of the Massachusetts Institute of Technology and Harvard University, Boston, MA, USA. [94]German Center for Diabetes Research (DZD e.V.), Neuherberg, Germany. [95]Department Psychiatry and Psychotherapy, University Medicine Greifswald, Greifswald, Germany. [96]Department of Clinical Phsiology and Nuclear Medicine, Kuopia University Hospital, Kuopio, Finland. [97]Department of Psychiatry, Amsterdam Neuroscience and Amsterdam Public Health Research Institute, Amsterdam UMC, Vrije Universiteit, Amsterdam, The Netherlands. [98]Public Health Sciences, Wake Forest School of Medicine, Winston-Salem, NC, USA. [99]SKLORG & Department of Epidemiology, Shanghai Cancer Institute, Renji Hospital, Shanghai Jiaotong University School of Medicine, Shanghai, P. R. China. [100]Dean's Office, University of Kentucky College of Public Health, Lexington, KS, USA. [101]Faculty of Medicine, University of Iceland, Reykjavik, Iceland. [102]Epidemiology, Tulane University School of Public Health and Tropical Medicine, New Orleans, LA, USA. [103]Sticht Center for Healthy Aging and Rehabilitation, Gerontology and Geriatric Medicine, Wake Forest School of Medicine, Winston-Salem, NC, USA. [104]The Charles Bronfman Institute for Personalized Medicine, The Icahn School of Medicine at Mount Sinai, New York, NY, USA. [105]The Mindich Child Health Development Institute, The Icahn School of Medicine at Mount Sinai, New York, NY, USA. [106]Cardiovascular Health Research Unit, Epidemiology, Medicine and Health Services, University of Washington, Seattle, WA, USA. [107]Kaiser Permanente Washington, Health Research Institute, Seattle, WA, USA. [108]Department of Population Quantitative and Health Sciences, Case Western Reserve University, Cleveland, OH, USA. [109]NHLBI Framingham Heart Study, Framingham, MA, USA. [110]Cardiology, Medicine, University of Mississippi Medical Center, Jackson, MS, USA. [111]Biostatistics, Preventive Medicine, University of Southern California, Los Angeles, CA, USA. [112]Department of Biostatistics, University of Washington, Seattle, WA, USA. [113]Clinical Pharmacology, William Harvey Research Institute, Barts and The London School of Medicine and Dentistry, Queen Mary University of London, London, UK. [114]NIHR Barts Cardiovascular Biomedical Research Centre, Queen Mary University of London, London, London, UK. [115]Division of Pulmonary Medicine, Department of Medicine, Beth Israel Deaconess Medical Center, Harvard Medical School, Boston, MA, USA. [116]These authors contributed equally: Raymond Noordam, Maxime M. Bos, Heming Wang, Thomas W. Winkler, Amy R. Bentley, Tuomas O. Kilpeläinen. [117]These authors jointly directed this work: Patricia B. Munroe, Ching-Ti Liu, Alanna C. Morrison, Dabeeru C. Rao, Diana van Heemst, Susan Redline. *email: r.noordam@lumc.nl; sredline@bwh.harvard.edu

