## [Peer Review File · Nature Communications]

Reviewers' Comments:

Reviewer #1:

Remarks to the Author:

Noordam and colleagues report on the interaction between extremes of total sleep time (both short total sleep time and long total sleep time, separately) and genome-wide single nucleotide polymorphisms (SNPs) in a multi-study multi-ethnic sample of over 120,000 individuals against 3 blood lipid traits. The authors used a 2 df test that jointly tests the SNP main effect and the interaction effect. They identified 59 novel SNPs related to lipid levels and further looked at these SNPs in other reported GWAS and with expression. The strengths of this study include the large sample size. Some clarifications and additional analyses would improve the manuscript.

Major comments:

1. The authors define novel loci as being > 1Mb distance from known loci, but there is no supplemental table with the list of 300+ SNPs used as known. What list was used as known loci? What is the evidence of interaction for all the known loci?
2. The authors report that no regions come up in both the long and short total sleep time analysis, but there appears to be some marginally significant associations in the opposite extreme results than the ones reported. I don't agree that there is no overlap. This seems like suggestive evidence.
3. It is interesting that some of the newly identified loci don't seem to have any LD friends. For example, rs7799249, long sleep time and HDL (supp Fig 1). What is happening here? Is this a poorly captured region for imputation? Or rs4075349, long sleep time and LDL, where there is substantial LD but significance drops substantially (supp Fig 3)?
4. While there is varying sample sizes in the different ethnic groups, do the effects remain consistent in direction in the various ethnic groups? Is there heterogeneity in the results due to including multiple ethnic groups?
5. No QQ plots are provided. What are the lambda GC values before the correction was applied and therefore used to correct the results?
6. The authors report proportion of variance explained estimates. Is this the proportion of variance explained by the interactions or the main effects of the novel SNPs, or both? The additional TG variance explained (4.25%) seems high given that these SNPs will have a smaller effect than the reported association and the number of reported SNPs reported here.
7. For the UK Biobank analyses, did you categorize sleep duration as was done in the discovery and replication analyses to make more comparable to that analysis? Could that effect the results?

Minor comments:

1. There are typos throughout the manuscript including on lines 364, 552-554, 554, and TG N in exposed in Fig 1.
2. Where ancestry groups self-defined or determined genetically? This should be clarified.
3. Line 680 – please specify the packages used in R for the linear mixed models. R is not specific enough.

Reviewer #2:

Remarks to the Author:

This manuscript investigates whether short and long sleep are associated with adverse lipid profiles. Potential lipid modification via behavioural interventions is an important topic, but one where more rigorous investigations are clearly required.

Particular strengths of this manuscript include it covering five different ancestry groups, and the presence of a replication dataset.

However, before I can recommend this manuscript for publication, there are some (relatively straightforward) clarifications I would like to see the authors make:

1) Why have the authors not used common definitions of <7hrs for short sleep, and >9hrs for long sleep? Given you are working with self-report data, it would make most sense to use these guidelines-based cutpoints.

2) In the discovery dataset why are the individuals classified as short sleepers (20.9%) and long sleepers (19.7%) not closer to 20% of the sample? 0.3% and 0.9% are fairly big deviations from the intended quintile-based cut-offs.

3) Why did you use a less stringent $5e-7$ threshold (rather than $5e-8$) in the discovery dataset?

4) In the look-ups and bioinformatics analyses, it would be much better to compare to summary stats that have published not only on the self-report, but also objective accelerometer-based sleep traits from UK Biobank. See recent Nature Communications/Genetics papers from Saxena&Weedon's team, plus Doherty&Lindgren's team.

5) It would be good to run a genetic correlation analysis between the long- and short- sleep results i.e. not just comparing the SNPS which are below the bonferroni p-value threshold, but looking at the entire signal. The Broad Institute's LDSC tool is easy to use for this.

6) The discussion section should more clearly acknowledge the limitations of using self-report measures of sleep. From memory the genetic correlation between self-report and accelerometer measures in UK Biobank is $r \sim 0.3$ (reported somewhere in the Saxena/Weedon/Doherty/Lindgren papers). There is quite a discrepancy in these measures, and that should be acknowledged.

7) In the methods section, you don't mention running HWE checks on the SNPS. I imagine you did this, but forgot to report it?

8) This is possibly out of scope, but given the recent availability of UK Biobank lipid data (released ~8 weeks ago), did the authors not think to include this in their analyses (and thus providing a huge boost in power)?

Reply to comments from the editor and reviewers on manuscript NCOMMS-19-07363A

We thank the reviewers for the critical and valuable comments on our manuscript. We also thank the editor for allowing to revise and resubmit our paper to Nature Communications. A point-by-point reply on the comments of the reviewer is presented below. All changes to the manuscript are highlighted in yellow.

Thank you for reconsidering our manuscript for publication in Nature Communications.

Sincerely,

On behalf of the co-authors,

Raymond Noordam PhD

Susan Redline MD, MPH

Reply to comments from reviewer 1:

Noordam and colleagues report on the interaction between extremes of total sleep time (both short total sleep time and long total sleep time, separately) and genome-wide single nucleotide polymorphisms (SNPs) in a multi-study multi-ethnic sample of over 120,000 individuals against 3 blood lipid traits. The authors used a 2 df test that jointly tests the SNP main effect and the interaction effect. They identified 59 novel SNPs related to lipid levels and further looked at these SNPs in other reported GWAS and with expression. The strengths of this study include the large sample size. Some clarifications and additional analyses would improve the manuscript.

Major comments:

COMMENT 1: The authors define novel loci as being > 1Mb distance from known loci, but there is no supplemental table with the list of 300+ SNPs used as known. What list was used as known loci? What is the evidence of interaction for all the known loci?

REPLY: We thank the reviewer for this comment. We now added the list of the 300+ SNPs that have been identified in previous studies and assessed the 1df interaction with either short or long total sleep time for these SNPs on the studied lipid levels (new supplementary tables 8 and 9). , We did not find evidence of a 1df interaction with any of the SNPs, although we found some evidence of SNPs that were located within 1 Mb from these variants. We added this information to the revised manuscript.

Changes made to the manuscript:

- *We added new supplementary tables in which we presented the 1df interactions by either LTST or STST with the known lead SNPs based on literature (PubMed IDs are provided in the new supplementary tables 8 and 9).*
- *Added to the results section: “In addition, we looked at the 427 known lipid SNPs (Supplementary Table 9), but these did not reveal significant 1df interactions with either LTST or STST.”*

COMMENT 2: The authors report that no regions come up in both the long and short total sleep time analysis, but there appears to be some marginally significant associations in the opposite extreme results than the ones reported. I don't agree that there is no overlap. This seems like suggestive evidence.

REPLY: The reviewer is correct that there are some genetic variants that show a significant joint effect for STST and both a suggestive joint effect for LTST, or vice versa. However, it is important to note that these variants did not come up when using the 1df interaction test. Therefore, these associations appear to predominantly reflect a SNP main effect on the blood lipid level rather than an interaction with either short or long total sleep time. In the revised manuscript, we emphasized this in the results section.

Changes made to the manuscript:

- *We added the following sentence to the results section: “Some of the novel SNPs identified through modelling a short or long sleep duration interaction (1 df) also showed suggestive evidence of association with lipid levels in the joint model (2 df test). However, this pattern suggested a main effect that appeared once sleep duration was adjusted for rather than an effect due to an interaction between sleep and the novel SNPs (Supplementary Table 10-11).”*

COMMENT 3: It is interesting that some of the newly identified loci don't seem to have any LD friends. For example, rs7799249, long sleep time and HDL (supp Fig 1). What is happening here? Is this a poorly captured region for imputation? Or rs4075349, long sleep time and LDL, where there is substantial LD but significance drops substantially (supp Fig 3)?

REPLY: Although lone signals can indicate spurious findings, we could not find any other evidence to invalidate these kind of loci. We assume that the fact that these signals are not represented by more variants is a consequence of the generally low LD in these regions (which was confirmed by HaploReg; for example, rs7799249 has, despite being a common SNPs, no neighboring SNPs with LD > 0.4 in any of the available ancestries). We were sensitive to potential issues with imputation quality throughout these analyses, and included the following filters: all variants must have an imputation quality of at least 50% and the product of the imputation quality and the minor allele count in both the exposed and unexposed strata had to be at least 20. Furthermore, in the case of rs7799249 the median imputation quality for was 92% and in the case of rs4075349 the median imputation quality was 98%. In the discussion section, we now highlighted this in more detail.

Changes made to the manuscript:

- *Added the following to the discussion section (page 22): “While for some of the novel SNPs had no neighboring SNPs in high LD (e.g., rs7799249; mapped to ATP6V0A4), our applied filters (e.g., imputation quality > 0.5) would suggest that the chance of invalidity of the findings is negligible. Furthermore, in the case of rs7799249, no SNPs in high LD are known in individuals from different ancestries.”*

COMMENT 4: While there is varying sample sizes in the different ethnic groups, do the effects remain consistent in direction in the various ethnic groups? Is there heterogeneity in the results due to including multiple ethnic groups?

REPLY: We agree with the reviewer that showing the consistency across ancestries could be of added value. In the revised manuscript, we added this information to the supplementary tables. Note that we did not added the Brazilian analyses given the very limited sample size.

Changes made to the manuscript:

- *Added the effect estimates of the SNP-main and SNP-interaction effect to the supplementary tables of the novel findings.*

COMMENT 5: No QQ plots are provided. What are the lambda GC values before the correction was applied and therefore used to correct the results?

REPLY: In line with the comments from the reviewer, we added the QQ plots of the overall meta-analyses to the supplementary materials. Furthermore, we added the lambda values of the meta-analyses before and after the second genomic control.

Changes made to the manuscript:

- *Added the following sentence to the results section: “QQ plots of the combined multi-ancestry and European meta-analysis of the discovery and replication analysis are presented in **Supplementary Figures 1 and 2**. Lambda values ranged between 1.023 and 1.055 (trans-ancestry meta-analysis) before the second genomic control and were all 1 after second genomic control correction.”*

COMMENT 6: The authors report proportion of variance explained estimates. Is this the proportion of variance explained by the interactions or the main effects of the novel SNPs, or both? The additional TG variance explained (4.25%) seems high given that these SNPs will have a smaller effect than the reported association and the number of reported SNPs reported here.

REPLY: The VarExp statistical package was used to calculate the variance explained by the combined SNP-main and SNP interaction effect (the joint effect). We agree with the reviewer that this was not sufficiently clear in the original submitted paper. In the revised manuscript, we added this explanation to the method section. We checked the script we used for the calculations with one of the developers of the script, and we confirm that calculations have been performed properly.

Changes made to the manuscript:

- *Added the following sentence to the methods section: “This package calculates the variance explained on the basis of the combined (joint) SNP-main and SNP-interaction effect.”*

COMMENT 7: For the UK Biobank analyses, did you categorize sleep duration as was done in the discovery and replication analyses to make more comparable to that analysis? Could that effect the results?

REPLY: The results on sleep variables in the UK Biobank were retrieved on the basis of publically available summary statistics. The sleep duration variable was considered continuously. In the revised manuscript, we emphasized this in the methods section.

Changes made to the manuscript:

- *We added the following sentence in the section in the methods on the follow-up analyses: “Analyses on these phenotypes were generally done using continuous outcomes; the variable “sleep duration” was expressed in hours of total sleep per day.”*

Minor comments:

COMMENT 8: There are typos throughout the manuscript including on lines 364, 552-554, 554, and TG N in exposed in Fig 1.

REPLY: We thank the reviewer for noticing these small errors. In the revised manuscript, we checked for all errors, and corrected them accordingly.

COMMENT 9: Where ancestry groups self-defined or determined genetically? This should be clarified.

REPLY: The different ancestries were self-defined. In the revised manuscript, we emphasized this in the methods section.

Changes made to the manuscript:

- *Changed the following sentence: Discovery and replication analyses comprised men and women between the age of 18 and 80 years, and were conducted separately for the different contributing (self-defined) ancestry groups, including: European, African, Asian, Hispanic, and Brazilian (discovery analysis only).”*

COMMENT 10: Line 680 – please specify the packages used in R for the linear mixed models. R is not specific enough.

REPLY: We thank the reviewer for noticing this missing information. These cohorts used the R package sandwich. We specified the R package in the revised manuscript.

Reply to comments from reviewer 2:

This manuscript investigates whether short and long sleep are associated with adverse lipid profiles. Potential lipid modification via behavioral interventions is an important topic, but one where more rigorous investigations are clearly required. Particular strengths of this manuscript include it covering five different ancestry groups, and the presence of a replication dataset.

However, before I can recommend this manuscript for publication, there are some (relatively straightforward) clarifications I would like to see the authors make:

COMMENT 1: Why have the authors not used common definitions of <7hrs for short sleep, and >9hrs for long sleep? Given you are working with self-report data, it would make most sense to use these guidelines-based cutpoints.

REPLY: We decided to use cohort-specific residuals, adjusted for age and sex, due to the wide differences in participant characteristics across cohorts and different instruments used to assess sleep duration. While it would be attractive to apply a simple, single threshold to define long and short sleep, there can be large differences in classification resulting from differences in how questions are asked. Moreover, there may be cultural biases in how individuals report sleep relative to gold standard sleep assessments (see Jackson CL et al, 2018, Sleep). In the revised manuscript, we expanded the text on this model in more detail to further explain our chosen strategy.

Changes made to the manuscript:

- *Changes the following sentence in the methods section: “For the present project, we defined both STST and LTST. To harmonize the sleep duration data across cohorts from different countries, cultures and participants with different physical characteristics, in whom sleep duration was assessed using various questions, we defined STST and LTST using cohort- specific residuals, adjusting for age and sex. An exception was for AGES and HANDLS....”*

COMMENT 2: In the discovery dataset why are the individuals classified as short sleepers (20.9%) and long sleepers (19.7%) not closer to 20% of the sample? 0.3% and 0.9% are fairly big deviations from the intended quintile-based cut-offs.

REPLY: In the case of the AGES and HANDLS studies, we were not able to calculate the age- and sex-adjusted residuals for total sleep duration as these cohorts had only a limited number of alternatives in the multiple-choice question. Only in these two cases, we used an alternative cutoff (and this even meant that HANDLS did not contribute to the LTST analysis) on the basis of the available data. Furthermore, age- and sex-adjusted residuals for sleep duration could end up being equal in the case that age (if with zero decimals), sex and sleep duration were the same. In the revised manuscript, we explicitly mentioned that residuals were not calculated for the AGES and HANDLS studies.

Changes made to the manuscript:

- *Changes the following sentence in the methods section: “An exception was for AGES and HANDLS cohorts, we used a cohort-specific definition due to limited response categories in relationship to the available question on sleep duration. Instead, we defined STST or LTST based on expert input.”*

COMMENT 3: Why did you use a less stringent 5e-7 threshold (rather than 5e-8) in the discovery dataset?

REPLY: Despite the large size of our discovery sample, the number of individuals in either the STST or LTST group was still limited. To limit false-negative results as much as possible, we decreased our threshold for the discovery analysis to 5e-7 instead of 5e-8. Please note that the novel SNPs had to have a p-value < 0.05 (with similar direction of effect) in the replication sample and a p-value < 5e-8 in the combined meta-analysis of the discovery and replication analysis. In the revised manuscript, we elaborated on this in the methods section.

Changes made to the manuscript:

- *Changed the following sentence in the methods section: “From this effort, we selected all SNPs associated with any of the lipid traits with $p \leq 5 \times 10^{-7}$ for replication in the Stage 2 analysis. This cut-off was selected to minimize false-negative results.”*

COMMENT 4: In the look-ups and bioinformatics analyses, it would be much better to compare to summary stats that have published not only on the self-report, but also objective accelerometer-based sleep traits from UK Biobank. See recent Nature Communications/Genetics papers from Saxena&Weedon's team, plus Doherty&Lindgren's team.

REPLY: We appreciate this suggestion from the reviewer. In line, we added this look-up to the revised manuscript.

Changes made to the manuscript:

- *Added the following sentence to the methods section: “We furthermore extracted the newly identified lead SNPs from the GWAS analyses done on accelerometer-based sleep variables, which was done in European-ancestry individuals from the UK Biobank (N = 85,670).”*
- *Added the following sentence to the results section: “Furthermore, we did not observe that any of the newly identified SNPs were associated with accelerometer-based sleep traits (Supplementary Table 18).*

COMMENT 5: It would be good to run a genetic correlation analysis between the long- and short- sleep results i.e. not just comparing the SNPS which are below the bonferroni p-value threshold, but looking at the entire signal. The Broad Institute's LDSC tool is easy to use for this.

REPLY: We appreciate the suggestion of the reviewer, and agree with the reviewer that such analyses would be of interest to further explore genetic effects on lipid levels in short vs long sleepers. In line with the suggestion of the reviewer, we explored two approaches. In one approach, we attempted to investigate the genetic correlations for the interaction estimates (using the 1d.f. test) on the three lipids. Unfortunately, the h^2 that is required for calculating the genetic correlation with LDSC is out of bounds in all cases, and therefore results were not given. Furthermore, given the overlap in the study populations, this could also have resulted in inflated p -values and false-positive findings. A second attempt was made by calculating the genetic correlation between the main effects on lipids stratifying by short and long sleeps (exposed to either STST or LTST), which also did not yield an estimate, likely due to the smaller sample size. We hope that future attempts in larger samples will be more successful.

COMMENT 6: The discussion section should more clearly acknowledge the limitations of using self-report measures of sleep. From memory the genetic correlation between self-report and accelerometer measures in UK Biobank is $r \approx -0.3$ (reported somewhere in the Saxena/Weedon/Doherty/Lindgren papers). There is quite a discrepancy in these measures, and that should be acknowledged.

REPLY: We agree with the reviewer that this recent finding should be discussed in the context of our paper. Please note that there are also phenotype correlation is only modest, which suggests that each approach characterizes somewhat different phenotypes (see Jackson et al, 2018, Sleep). In the revised manuscript, we further elaborated on this.

Changes made to the manuscript:

- *Added the following sentence to the discussion section: “.. , although the overall genetic correlation with accelerometer-based sleep duration was shown to be low. Moreover, observational studies showed only a modest correlation between the phenotypes, which suggest that each approach characterizes somewhat different phenotypes”*

COMMENT 7: In the methods section, you don't mention running HWE checks on the SNPS. I imagine you did this, but forgot to report it?

REPLY: The reviewer is correct that the genotyped data has been filtered for SNPs not in HWE on a cohort level.

Changes made to the manuscript:

- *We added the following information to the paragraph on the genotype data in the methods section: “Each study conducted imputation using various software programs and with local cleaning thresholds for call rates (usually > 98%) and Hardy-Weinberg equilibrium (usually p -value < $1e-5$).”*

COMMENT 8: This is possibly out of scope, but given the recent availability of UK Biobank lipid data (released ~8 weeks ago), did the authors not think to include this in their analyses (and thus providing a huge boost in power)?

REPLY: We thank the reviewer for this comment. As the reviewer most likely knows the release of the biomarker data from the UK Biobank was postponed several times throughout the past year. Given our already quite large number of new loci, we therefore decided to move forward instead of waiting. We will however continue our projects and hopefully incorporate the UK Biobank (as well as other additional samples) in the future.

Reviewers' Comments:

Reviewer #1:

Remarks to the Author:

The authors have adequately addressed my previous comments and I do not have any additional comments.

Reviewer #2:

Remarks to the Author:

I have very closely read through the revised manuscript and am satisfied that the authors have appropriately addressed each of the original issues I raised - this is a thorough revision.

I would like to request two minor edits:

1) The second last paragraph of the discussion section should more clearly acknowledge that larger studies are needed in future i.e. pointing towards the release of the biomarker data from the UK Biobank.

2) Update the reference list to update details of bioRxiv citations which have now been published.

Reply to comments from the editor and reviewers on manuscript NCOMMS-19-07363B

We are delighted that our manuscript is, in principal, accepted for publication in Nature Communications. A point-by-point reply on the comments of the editorial office and reviewer 2 is presented below. All changes to the manuscript are highlighted in yellow.

Thank you for reconsidering our manuscript for publication in Nature Communications.